# An Overview of Circulating Biomarkers in Neuroendocrine Neoplasms: A Clinical Guide

**DOI:** 10.3390/diagnostics13172820

**Published:** 2023-08-31

**Authors:** Michele Bevere, Francesca Masetto, Maria Elena Carazzolo, Alice Bettega, Anastasios Gkountakos, Aldo Scarpa, Michele Simbolo

**Affiliations:** 1ARC-Net Research Center, University of Verona, 37134 Verona, Italy; michele.bevere@univr.it (M.B.); francesca.masetto@univr.it (F.M.); anastasios.gkountakos@univr.it (A.G.); aldo.scarpa@univr.it (A.S.); 2Department of Diagnostics and Public Health, Section of Pathology, University and Hospital Trust of Verona, 37134 Verona, Italy; mariaelena.carazzolo@studenti.univr.it (M.E.C.); alice.bettega@studenti.univr.it (A.B.)

**Keywords:** neuroendocrine neoplasms, circulating biomarkers, liquid biopsy, mono-analyte biomarkers, multi-analyte biomarkers, diagnostic biomarkers, predictive biomarkers of treatment response, disease monitoring biomarkers

## Abstract

Neuroendocrine neoplasms (NENs) are a heterogeneous group of diseases that are characterized by different behavior and clinical manifestations. The diagnosis and management of this group of tumors are challenging due to tumor complexity and lack of precise and widely validated biomarkers. Indeed, the current circulating mono-analyte biomarkers (such as chromogranin A) are ineffective in describing such complex tumors due to their poor sensitivity and specificity. In contrast, multi-analytical circulating biomarkers (including NETest) are emerging as more effective tools to determine the real-time profile of the disease, both in terms of accurate diagnosis and effective treatment. In this review, we will analyze the capabilities and limitations of different circulating biomarkers focusing on three relevant questions: (1) accurate and early diagnosis; (2) monitoring of disease progression and response to therapy; and (3) detection of early relapse.

## 1. Introduction

Neuroendocrine neoplasms (NENs) are a heterogeneous class of rare tumors that can virtually arise from every part of the body with different histopathological, molecular, and clinical features [1]. The incidence of NENs account for 0.5% of all malignancies [2] and metastases are present in 21% to 69% of patients due to delayed and challenging diagnosis [3].

The current WHO classification [4] divides NENs into: well-differentiated (also known as neuroendocrine tumors (NETs); 80–90%), poorly differentiated (also known as neuroendocrine carcinoma (NEC); 10–20%) and mixed neuroendocrine/non-neuroendocrine form (also known as MiNEN). NETs are further subclassified according to mitotic count and Ki67 index in G1 (<2 mitotic count/mm^2^; <3% Ki67 index), G2 (2–20 mitotic count/mm^2^; 3–20% Ki67 index) and G3 (>20 mitotic count/mm^2^ and >20% Ki67 index). On the contrary, NECs are defined by a mitotic count and Ki67 index of >20 and divided in two different morphologies: large and small cells [4].

Generally, NETs are indolent malignancies associated with slow progression, whereas NECs are aggressive tumors with higher proliferation and metastasis rate [5,6]. Furthermore, NETs can be divided into functioning and non-functioning tumors. The functioning NETs secrete excessive amounts of hormones causing different associated-syndromes and have a better prognosis; conversely, the non-functioning NETs do not release hormones and are associated with poor outcomes [5,6]. The available diagnostic tools include Ki67, and immunohistochemistry for chromogranin A (CgA), synaptophysin, CDX2, protein gene product 9.5 (PGP 9.5), CD56, and thyroid transcription factor 1 (TTF-1) combined with standard diagnostic tools; in the case of functioning NENs, detection of serotonin, gastrin, and other hormones are mandatory to discriminate the various subclasses [7]. Additionally, in order to discriminate between NET G3 and NEC the current European guidelines recommend molecular analysis to test the alteration status of *MEN1/ATRX/DAXX* and *RB1/TP53* genes [8,9].

Overall, the diagnosis of NENs is still challenging due to their heterogeneity, different morphogenic and clinical features, as well as the absence of widely available circulating biomarkers. Due to the invasiveness of biopsy and the limitations of histopathology, there is an urgent need for non-invasive and reproducible biomarkers. In line with this, the multi-analyte circulating biomarkers are demonstrating promising advantages in the field of NENs.

This state-of-the-art review aims to summarize the main circulating biomarkers with an impact on the clinical routine.

For that, the search strategy on PubMed included combinations of the following keywords: #1 “circulating biomarkers”[Title/Abstract] AND “neuroendocrine tumors”[Title/Abstract]; #2 “circulating biomarkers”[Title/Abstract] AND “neuroendocrine carcinomas”[Title/Abstract]; #3 “circulating biomarkers”[Title/Abstract] AND “diagnostic tool”[Title/Abstract] AND “NETs”[Title/Abstract]; OR “NECs”[Title/Abstract]; OR “NENs”; #4 “circulating biomarkers”[Title/Abstract] AND “monitoring disease”[Title/Abstract] AND “NETs”[Title/Abstract]; OR “NECs”[Title/Abstract]; OR “NENs”; #5 “circulating biomarkers”[Title/Abstract] AND “early relapse”[Title/Abstract] AND “NETs”[Title/Abstract]; OR “NECs”[Title/Abstract]; OR “NENs”. No systematic search/review of the literature was performed.

## 2. Circulating Biomarkers in NENs: Mono-Analytes versus Multi-Analytes

Mono-analyte biomarkers are specific molecules (i.e., CgA, circulating tumor cells, and serotonin) detectable in the blood or other body fluids used to diagnose tumors, detect the presence of disease, and monitor tumor progression. The main limitations of mono-analyte biomarkers include the high heterogeneity of NENs, the absence of a standardized method of analysis, and the lack of secretory products in most patients affected by NENs [10,11]. To overcome the limitations of mono-analyte biomarkers, several multiple-analyte biomarkers (i.e., NETest, microRNA, and circulating tumor DNA) are under investigation in the field of NENs, with NETest showing the most promising results [10,11].

## 3. Circulating Biomarkers in NENs

A promising alternative for rapid and minimally invasive molecular diagnostics is the liquid biopsy (Figure 1). This technique would allow the analysis of tumor-derived circulating elements in the body fluids for monitoring tumor evolution at different stages [12,13]. 

In the next chapters, a thorough overview focusing on circulating biomarkers, both mono-analytes and multi-analytes, with clinical relevance in the field of NENs is presented.

### 3.1. Available Mono-Analyte Circulating Biomarkers

#### 3.1.1. CgA

CgA is an acid glycoprotein stored in the secretory granules of most endocrine and neuroendocrine cells, where it is released together with peptide hormones and biogenic amines [14]. Circulating CgA has been correlated with tumor burden, progression, and metastasis [15]. Thus, it represents a broad-spectrum marker for NENs (Table 1). However, its use in the clinic is hampered by issues that compromise both specificity and sensitivity. Indeed, increased CgA expression may not be due to the presence of NENs, but may be affected by a number of NEN-independent conditions, both benign and malignant in origin [16,17]. On the other hand, intrinsic features of neuroendocrine disease also correlate with high variability in CgA values and may lead to false positive results (Table 1). For instance, CgA levels vary according to tumor function and location, appearing to be higher in well-differentiated than in poorly differentiated, functional than in non-functional, and metastatic than in locoregional disease [18]. Noteworthily, 30–50% of patients with NEN do not show have increased CgA levels [19]. Another critical aspect is the lack of standardization of tests. For example, several commercial kits can measure CgA in serum or plasma and are based on various molecular affinity techniques (enzyme-linked immunosorbent assays, ELISA; radioimmunoassay assays, IRMA; and time-resolved amplified cryptate emission, TRACE) and different antibodies recognizing the full-length protein, fragments, or its derivatives [17,20]. These limitations occur evident when comparing studies that exclude or include patients with confounding factors (i.e., non-oncological conditions, assumption of proton-pump inhibitors, and non-NEN tumors), where specificity drops from about 90% to 60–50% [17]. However, CgA correlates with tumor function, degree of differentiation, and extent of disease. Its sensitivity is considered acceptable for well-differentiated NENs, but extremely poor for non-functioning localized tumors where CgA production is lower [20]. Overall, circulating CgA is considered of controversial value in diagnostic decision-making [21].

Regarding monitoring the disease, CgA is reported to be the most commonly used biomarker to assess the disease burden and monitor treatment response in NENs [22]. However, the available evidence on this role of CgA is controversial and limited by the small number of studies [17,22,23]. Through sub-analyses of the RADIANT-2 and -3 clinical trials, it was shown that baseline CgA levels do not predict the impact of therapy (everolimus vs. placebo) on the survival of patients with gastroenteropancreatic (GEP)-NET [17,24,25]. The same clinical trial, however, showed that early responders had a longer progression-free survival (PFS), compared with nonresponders [26]. Similarly, an association between decreased CgA and reduced risk of disease progression was seen in patients with GEP-NET treated with lanreotide [27,28]. In contrast, on a cohort of patients with GEP- and bronchopulmonary-NENs treated with peptide receptor radionuclide therapy (PRRT), CgA failed to reflect the disease course [29]. Interestingly, different studies have revealed its role in predicting disease progression [30,31,32], especially for advanced NENs and gastrinomas [33,34,35]. However, some limitations have been revealed [17,27,28,29,36,37,38,39]; thus, current guidelines advise that treatment decisions should not be based only on CgA results [40].

Few studies interrogated the role of CgA to predict tumor recurrence. In pancreatic NETs, several studies showed that CgA is a good tool to predict tumor relapse after surgery [34,35,36,41,42,43,44]; whereas, other studies concluded that CgA has a limited value [45]. Moreover, the application of CgA is not sufficient to predict tumor relapse for medullary thyroid NETs and lung NENs [46,47]. 

#### 3.1.2. Circulating Tumor Cells 

Circulating tumor cells (CTCs) are tumoral cells considered as metastatic precursors [12,48] which are associated to worse PFS and overall survival (OS) in different solid tumors [48,49]. These cells detach as individuals or groups from the original solid tumor, enter into blood vessels, and through the bloodstream reach a distant site to take root and give rise to a secondary tumor [12,50]. Although CTCs have become of interest in several solid tumors [48], they are still under investigation for NENs (Table 1) [51]. 

The only FDA-approved method for the isolation and enumeration of CTCs from blood is the CellSearch^®^ (Janssen Diagnostic, Beerse, Belgium) system, which is based on the immunocapture of CTCs with antibody anti-epithelial adhesion molecules (EpCAM) [52]. This method implies the expression of EpCAMs by CTCs but may be a problem because cells can undergo epithelial-to-mesenchymal transition, thereby leading to a phenotype change with the downregulation of epithelial markers and upregulation of the mesenchymal ones [12,50]. Other enrichment methodologies based on density and size are represented by NanoVelcro Chips [12]. The NanoVelcro Chips allow a better capture of CTCs through an increase in the contact surface area between anti-EpCAM-coated nanostructured substrates and cells surface components [53,54]. However, in patients with high-grade NETs, the number of CTCs in the blood is higher compared to low-grade NETs that have a slower pattern [55], thus requiring higher sensitivity for their detection. The presence of CTCs in the blood correlates with increased tumor burden, grade, and serum CgA levels [55,56]. Although counting CTCs may be of interest in patients with NENs, their diagnostic usefulness is rather low for the difficulty in identifying and isolating them accurately [11]. Indeed, CTCs are present in less than 50% of patients and therefore do not provide adequate diagnostic accuracy [21,57]. Moreover, CTCs are mono-analyte markers that may not be fully representative of the tumor because they are derived from a portion of the tumor and/or may undergo subclonal alterations [21,58]. To date, there is no robust enough evidence for the use of CTCs as diagnostic tools in NENs.

CTCs detection can be applied during the monitoring of treatment [59,60]. In a first study, in pancreatic NET, intestinal NET, and NET of unknown origin, CTCs positive for somatostatin receptor (SSTR) were associated with lower tumor grade than those without SSTR, partly explaining the escape from disease control in patients treated with somatostatin analogue (SSA) or PRRT and thus giving insights on the choice of therapy. The subsequent CALM-NET clinical trial evaluated the potential use of CTCs to monitor disease progression. This study concluded that the absence of CTCs at baseline correlated with a higher change in symptomatic response to treatment in patients with midgut NET [61]. Furthermore, in patients with metastatic nonfunctioning midgut and bronchopulmonary NET, an undetectable or 50% decrease in CTCs from baseline post-therapy was associated with a reduced likelihood of disease progression [62]. In addition to the levels of CTCs, analysis of copy number alterations (CNAs) of CTCs could also be an additional predictive marker. The authors of a recent report on patients with small cell lung NEC identified a profile of CNAs in CTCs that can correctly distinguish chemo-refractory and chemo-sensitive patients [13,63]. A high level of CTCs is associated to worse PFS and OS in NENs [49,55,64,65,66,67,68]; whereas the reduction in CTCs after treatments is associated with a better PFS and OS in patients with NENs [60]. Recently, the NanoVelcro Chip assay in patients with advanced NET undergoing PRRT detected dynamic changes in the number of single, clustered, and total CTCs strongly associated with treatment responses [68]. 

Few studies investigated the function of CTCs to foretell tumor relapse, showing that the increase in CTC levels may predict metastasis formation in patients with NET [62,68].

#### 3.1.3. Other Biomarkers 

Additional general biomarkers used in NENs are neuron-specific enolase (NSE), pancreatic polypeptide (PP), and neuropeptide Y (NPY) [69].

NSE is an isoform of the glycolytic enzyme enolase present in neurons and neuroendocrine cells, and its assessment can give insight into the altered metabolism or turnover of these cells [70]. It is recognized as the first-choice biomarker for the diagnosis of bronchopulmonary NETs, in particular small cell lung cancer [71] because its sensitivity and specificity are higher compared to other biomarkers, such as CgA [38], when considering high-grade and poorly differentiated tumors [20,72]. However, the sensitivity of NSE alone is not very high; therefore, a combination with other biomarkers, such as CgA, may be a valid option [59,73]. Moreover, the available methods to detect circulating NSE are several, including ELISA, electro-chemiluminescence immunoassays (ECSIA), and radioimmunoassay (RIA), raise concerns about the measurement reliability depending on the assay of choice [74].

PP is a 36-amino-acid molecule expressed by endocrine cells of the colon and pancreas involved in the regulation of the digestive tract function and food metabolism [75]. PP sensitivity is quite low (63% in pancreatic NETs and 53% in gastrointestinal NETs) and a poor correlation is found between the change in PP in serum and radiological imaging [76]. However, in Sansone et al., the combination of PP with CgA resulted in an increased sensitivity, mainly for non-functioning pancreatic NETs [69,77].

NPY family, to which PP belongs, is a group of three homologous peptides with different functions but all with pro-tumoral effects [75,78]. NPY is a neurotransmitter whose high plasma levels are found in several cancers, including pheochromocytoma, ganglioma, and neuroblastoma, in which it can be used as a marker [59,78]. Despite the measurement of catecholamines displaying a higher sensitivity in pheochromocytoma and paraganglioma, NPY may be a viable alternative in patients suffering from kidney impairment or under treatments that interfere with catecholamine reuptake [79]. However, its clinical use is limited due to the low amount of information available, thus requiring further in-depth studies [79]. 

Moreover, there are other tumor-specific biomarkers used for the diagnosis of different functioning NENs (Table 2).

Different biomarkers are used to more specifically diagnose subtypes of functioning pancreatic NENs in combination with other diagnostic (CgA) and imaging tools [69,80]. These markers include insulin, somatostatin, glucagon, gastrin, vasoactive intestinal peptide (VIP), serotonin, adrenocorticotropic hormone (ACTH), catecholamines, calcitonin, growth factor, insulin growth factor 1, and prolactin.

Insulin is a peptide hormone secreted by the beta cells of pancreatic islets as a response to high blood glucose levels. High levels of insulin are associated with insulinoma causing the Whipple’s triad (hypoglycemia, low plasma levels of glucose, and resolution of symptoms after correction of the hypoglycemia) [81]. Once a hypoglycemic event is confirmed (glucose levels ≤ 2.1 mmol/L), the levels of insulin and pro-insulin must be monitored during a supervised 48–72-h fast [82]. At the end of fasting, an insulin concentration ≥ 5 μIU/mL and a proinsulin concentration > 22 pmol/L represent the cutoffs for the diagnosis of insulinoma [83,84]. However, an increase in insulin and proinsulin may also be due to non-neoplastic conditions, such as early morning pre-prandial or after exercise [16].

Somatostatin is a hormone secreted by pancreatic delta cells and gastric antrum D cells [12]. When secreted in excess it is associated with somatostatinoma, thereby causing the classic triad of somatostatinoma syndrome (diabetes/glucose intolerance, cholelithiasis, and diarrhea/steatorrhea) [81]. Given the rare incidence of these rare functioning pancreatic NENs, serum somatostatin levels should be measured only in the presence of somatostatinoma syndrome.

Glucagon is a peptide hormone produced by the alpha cells of the pancreas when glucose levels are low [12]. High fasting levels of glucagon are associated with the diagnosis of glucagonoma leading to a typical triad of glucagonoma syndrome (skin rash, diabetes mellitus, and weight loss). However, high levels of glucagon can also be found in different non-neoplastic conditions, such as cirrhosis, diabetes mellitus, sepsis, and burns [59].

Gastrin is a peptide hormone implied in chloride acid release from parietal cells of the stomach, gastric motility, and pancreatic secretion [12]. Excessive production of gastrin during fasting combined with increased gastric acid output leads to gastrinomas and Zollinger–Ellison syndrome (duodenal ulcer and/or gastro-esophageal reflux disease) [85]. However, high levels of gastrin can be found in several non-neoplastic conditions, such as atrophic gastritis, Helicobacter pylori infection, or proton-pump inhibitor treatment [86]. Indeed, the fasting gastrin test should be performed in the presence of gastric acid hypersecretion (pH ≤ 2) without the interference of proton pump inhibitors. In patients under proton pump inhibitor treatment, it is advised to switch to histamine type 2 receptor blockers for 1–2 weeks before the gastrin measurement and antacids for 1–2 days before the test [85].

VIP is a hormone released by pancreatic and brain cells that promotes vasodilation, regulates smooth muscle activity, and inhibits gastric acid secretion [12]. Excessive VIP secretion (>60 pmol/L) combined with diarrhea is related to VIPoma with the Verner–Morrison syndrome (diarrhea, hypokalemia, hypochlorhydria/achlorhydria, and acidosis) [87,88]. However, mild levels of serum VIP can occur in other non-neoplastic conditions, such as short bowel syndrome and inflammatory diseases [89].

Serotonin and its main metabolite, 5-hydroxyindoleacetic acid (5-HIAA), are assessed in patients with NEN showing carcinoid syndrome (abdominal pain, diarrhea, weight loss, and flushing) [21]. The most performed assay is the 24 h urinary 5-HIAA measurement in patients with midgut NENs [16]. However, there are some false positives due to non-neoplastic conditions, including the dietary assumption of tryptophan-rich food and certain medications (e.g., diazepam and phenobarbital), which alter the serotine production [59]. Another limitation of this urinary assay is the urine collection for 24 h in which the patients collect all the urine produced over time. An alternative test is the 24 h serum 5-HIAA measurement. Given most circulating serotonin is stored in platelets, the serum 5-HIAA assay shows a higher sensitivity than the urinary 5-HIAA test. Moreover, the latter first assay is not influenced by diet [90]. Importantly, a complication of carcinoid syndrome may be carcinoid heart disease due to high levels of circulating vasoactive substances such as serotonin, tachykinins, and prostaglandins. Carcinoid heart disease is characterized by the thickening of cardiac valves and arrhythmias [91]. For this, the plasma levels of the amino-terminal pro-brain natriuretic peptide should be measured in addition to the serum 5-HIAA assay to diagnose carcinoid heart disease [92].

In addition to somatostatin, bronchial and pancreatic NENs also cause elevated ACTH levels, resulting in increased glucocorticoid levels that cause Cushing’s syndrome [93].

Catecholamines (CAs), which include dopamine, norepinephrine, and epinephrine, are neurotransmitters and hormones essential for maintaining homeostasis via the autonomic nervous system [94]. The pathological increase (in urine or plasma) of metanephrines, which are metabolites of CAs, is a highly sensitive screening test for pheochromocytomas and paragangliomas. These NENs originate in the adrenal medulla and the extra-adrenal autonomic paraganglia, respectively, but are classified together as paragangliomas by the WHO [95,96]. To improve diagnostic sensitivity and avoid false positive results, the test should be performed using chromatographic methods, taking into consideration the definition of age-related cut-offs [97]. In addition, sympathomimetic substances, including caffeine, nicotine, and various drugs, can interfere with the production of norepinephrine and epinephrine, leading to false results [95]. 

Calcitonin (CT) is a polypeptide hormone produced by parafollicular C cells located mainly in the thyroid gland, but also in other organs, including lungs, liver, pancreas, thymus, and small intestine. Elevated serum levels of CT and its precursor, procalcitonin (PCT), are strong indicators of medullary thyroid carcinoma [98,99]. However, it is worth considering that there is an extremely rare non-secretory form of medullary thyroid carcinoma and the increased levels of CT may be caused by non-neoplastic conditions, including renal failure and hyperparathyroidism [100,101,102]. In addition, CT measurement is characterized by multiple assays and protocols and different cut-offs that contribute to false positives [102,103]. Contrarily, CT (calcitonin doubling times and the more normalized postoperative calcitonin-to-preoperative calcitonin ratio) and PCT evaluations are considered strong prognostic markers in follow-up to assess medullary thyroid carcinoma recurrence [104,105,106,107].

An increase in plasma growth hormone (GH), insulin growth factor 1 (IGF1), prolactin, or cortisol (Cushing’s disease), on the other hand, is indicative of pituitary NET [108].

Finally, other possible biomarkers studied for GEP-NETs are circulating angiogenic molecules. Of these, VEGF, although the most powerful, is the most debated due to its highly controversial results. On the other hand, placental growth factor, angiopoietin 2, and IL-8 were found to be good predictors of unfavorable outcomes and aggressive disease behavior. However, there is currently no evidence to use them as routine markers in the clinic [109].

**Table 1 diagnostics-13-02820-t001:** Strengths and flaws of mono-analyte biomarkers in early diagnosis, monitoring response to therapy, and early detection of tumor relapse.

Mono-Analyte Biomarkers	Strengths	Flaws
CgA	Acceptable sensitivity only for well-differentiated NEN [20]	30–50% of false negative in patients with NEN [19]
	Associated to a longer PFS in GEP-NET (RADIANT-2 clinical trial) [26]	Non-standardized method of analysis [17,59]
	Marker of disease progression in advanced NENs and gastrinomas [33,34,35]	Poor specificity in NEN and poor sensitivity for non-functioning localized NET [20]
		Not effective in monitoring the disease in: GEP-NET treated with Everolimus (RADIANT-2 and 3 clinical trials) and GEP- and bp NENs treated with PRRT [17,25,110]
		Not effective in monitoring tumor relapse in medullary thyroid NETs and lung NENs [46,47,111]
CTC	Correlation between amount and treatment response in midgut NET (CALM-NET trial) [61]	EpCAM expression required for isolation method FDA-approved [52]
	Correlation between amount and disease progression in post-therapy metastatic nonfunctioning midgut and bp NET [62]	Detectable in less than 50% NENs [21,57]
	Analysis of CTC-derived CNAs identify chemo-refractory and chemo-sensitive SCL NECs [63]	Low levels detectable in low-grade NETs [54]
	Correlation between amount and PFS and OS in metastatic NENs [60]	
	Correlation between amount and metastasis formation in NENs [62,112]	

CgA = chromogranin A; CTCs = circulating tumor cells; GEP = gastroenteropancreatic; PFS = progression-free survival; OS = overall survival; CNAs = copy number alterations; bp NET = bronchopulmonary NET; PRRT = peptide receptor radionuclide therapy; SCL NEC = small cell lung NEC.

**Table 2 diagnostics-13-02820-t002:** Summary of tumor-specific biomarkers used in the diagnosis of different functioning NENs.

Type of Functioning NEN	Secreted Hormones
Pancreatic NENs	Insulin
	Glucagon
	Somatostatin
	Gastrin
	Vasoactive intestinal polypeptide (VIP)
	Adrenocorticotropic hormone (ACTH)
Gastrointestinal NENs	Serotonin
	Gastrin
	Glucagon
Lung NENs	Serotonin
	Adrenocorticotropic hormone (ACTH)
Pheochromocytoma and paraganglioma	Catecholamines (CAs) and metabolites
Thyroid NENs	Calcitonin (CT)
Pituitary NENs	Growth hormone (GH)
	Prolactin
	Insulin growth factor 1 (IGF1)
	Cortisol

NENs = neuroendocrine neoplasms.

### 3.2. Potential Novel Multi-Analytes Biomarkers for NENs

#### 3.2.1. NETest

NETest is a tool based on real-time PCR combined with deep learning strategies to specifically identify tumors with a neuroendocrine genotype [72,73,74].

mRNA is isolated from EDTA-collected whole blood samples and real-time PCR is performed to interrogate 51 genes with the aid of four different prediction algorithms [113]. The choice of these 51 genes was developed on tissue-based, blood-based, and literature-curated panels of genes in order to define the expression profile of NENs [80,113]. In addition, these genes have been confirmed as bona fide neuroendocrine markers in a large dataset (11,232 samples) from The Cancer Genome Atlas (TCGA) [114]. Results are expressed as a NET score which ranges from 0 to 100%. This score is directly proportional to the level of disease activity at the time of testing: 0–40% indicates low activity and is a sign of tumor stability, 41–79% and ≥80% correspond to moderate or high activity and are correlated with tumor progression.

Since its development, the NETest has been repeatedly documented to be a useful tool for detecting the presence of different types of NEN of different origins, including pancreas, lungs, small intestine, thymus, and even those of unknown origin, with an accuracy of more than 90% regardless of the stage or grade of the tumor (Table 3) [115]. 

A recent multicenter study in a cohort of three different types of NETs (GEP, bronchopulmonary, and of unknown origin) demonstrated the ability of NETest to discriminate NETs from a complex set of controls: healthy, non-NET malignancies, and benign diseases affecting CgA levels [114]. In this study, the diagnostic accuracy of both NETest and CgA was also compared, revealing the better performance of NETest (>91% vs. <50%, respectively) [114]. This high diagnostic accuracy of NETest was further confirmed in a meta-analysis of six different studies [116]. In contrast, a large independent validation study showed that NETest is more sensitive but less specific than CgA in GEP-NETs concluding that this precludes its use as a screening marker [117]. The authors hypothesized that the low specificity may be due to both a possible interference of gene expression caused by nonmalignant conditions as demonstrated also in another validation study on GEP-NET [118], and the presence of platelets and extracellular RNA in the source of the transcripts [117]. 

NETest has revealed promising results in monitoring of disease to differentiate stable from progressive disease in different subtypes of NENs including pulmonary, thymic, and GEP NETs [72]. Overall, studies agree that a NETest score > 40% is associated with disease progression in concordance with radiological imaging and also in accordance with a previously reported meta-analysis [72,116]. Furthermore, although the available data are few and difficult to compare, NETest might be able to predict tumor response under treatment [72]. A recent study defined the NETest as useful in guiding treatment strategy in combined with imaging [119]. Indeed, in a recent study, patients with GEP-, bronchopulmonary NET or of unknown origin, treated with SSA or other therapies having a baseline NET score of >80% were assessed as non-responders, while a <40% score was associated with responders. Low-score tumors supported no change in management, thereby reducing the need for imaging. Whereas a high score indicated the need for intervention and changes in treatment [119]. However, although NETest is a promising marker for treatment monitoring, the cut-off values to distinguish stable from progressive disease have not been standardized and vary among different studies [72]. In another recent validation study on GEP-NET treated with SSAs, everolimus or CAPTEM (capecitabine and temozolomide), NETest (cut-off 33%) reliably predicted stable disease and was the strongest predictor of progressive disease compared to CgA [120]. This predictive ability was also confirmed in GEP- and bronchopulmonary NETs treated with PRRT [121]. In this study, on one hand, a decreased NETest score identifies responsive tumors correlating with the independent biomarker PRRT predictive quotient (PPQ), which integrates blood-derived NET-specific gene transcripts and tissue Ki67 values. On the other hand, NETest readily identified non-responders in advance of currently used imaging methods [121]. 

Finally, the NETest was able to predict tumor recurrence with 94% accuracy compared with CgA after surgery [122]. In patients affected by pancreatic NETs, a decrease in the NETest score correlated with better surgical efficacy. In patients with small intestinal NETs, NETest revealed a strong tool to predict disease progression after surgery with a sensitivity of 100% and specificity of 77.78% [123]. These findings have been confirmed in a multicenter study with a higher cohort of patients with different NETs [123]. In a retrospective analysis, the NETest was demonstrated to be useful in detecting residual disease after surgery with >90% accuracy [124]. A recent study investigated blood samples of patients with GEP-NETs and healthy volunteers using both NETest and CgA [117]. The NETest sensitivity and specificity were 93% and 56%, while for CgA were 56% and 83%, respectively. This study revealed that the NETest showed a higher sensitivity but lower specificity than the CgA in the detection of residual disease after surgery [117]. Another study analyzed patients affected by GEP-NET with both CgA and NETest [125]. Positive results were also found in patients with GEP-NETs treated with SSAs [126]. The NETest was more accurate (96%) than CgA changes (around 25%) in predicting disease alterations over 5 years [125]. In patients affected by small bowel NENs at stage IV, the NETest score was higher prior to the treatment (surgery and PRRT) and decreased in accordance with tumor reduction after treatment [127]. Additionally, NETest showed a better correlation with other clinical parameters (i.e., imaging, tumor grade, Ki67 index) compared with CgA with an accuracy of >91% versus <50%, respectively [124]. These data were further confirmed by other independent studies [117,119,120]. Noteworthy, the NETest detected early liver metastasis in a patient with NET of the ileocecal valve, whereas the conventional biomarkers/imaging remained unaltered [128]. Overall, these data suggest that the NETest may be more accurate in detecting early relapses in NENs compared to available biomarkers [17,72,116,129]. 

#### 3.2.2. MicroRNAs 

MicroRNAs (miRNAs) are 21–25 nucleotide small non-coding RNAs, which act at the post-translational level by binding target RNAs to negatively regulate their expression [130]. miRNA can be found in tissues and/or released in body fluids in free form or in microvesicles (plasma, serum, urine, saliva, and cerebrospinal fluid) as a result of tissue injury, apoptosis, and necrosis [131]. miRNAs can be used as markers due to their abundance, specificity for cell type and disease stage, and stability. These aspects can be very advantageous in the diagnosis of NENs, both to distinguish poorly differentiated NETs from non-neuroendocrine tumors and to identify different molecular subgroups [132,133,134]. However, little is known about circulating miRNAs in NENs (Table 3), due to the lack of standardized analysis methods and inconsistency between tissue and circulating signatures [21,59,80,135,136]. Moreover, necrosis in G1 and G2 NET is uncommon; thus, these tumors do not represent an adapted source of miRNAs [134,137]. 

Regarding diagnostic capacity, five miRNAs were reported able to discriminate NETs from pancreatic ductal adenocarcinoma (PDAC) [138,139]. Among these five, miRNA-1290 had the best diagnostic performance. In addition, circulating miRNA-21 can differentiate the diagnosis of pancreatic NET from chronic pancreatitis [138]. Other studies showed that the overexpression of miRNA-1290 may discriminate PDAC from pancreatic NETs, whereas miRNA-584, -1285, -550a-5p, and -1825 are downregulated [139]. In the serum of patients with pancreatic NETs with MEN1 syndrome, miRNA-3156-5p is significantly downregulated compared to the control patients [140]. In small bowel NETs, a serum combination of four miRNAs (miRNA-125b-5p, miRNA-362-5p, miRNA-425-5p, and miRNA-500a-5p) was found to be able to differentiate NET from hepatocarcinoma. 

Few data are available on the potential correlation of circulating miRNAs with treatment status. It has been demonstrated that miRNA-222 is increased in patients with gastric NET and decreased after CCK2R antagonist netazepide (YF476) [141]. In well-differentiated small intestinal NETs, five miRNAs (miRNA-96, -182, -183, -196a, and -200a) are found to be upregulated during tumor progression, conversely, four miRNAs (miRNA-31, -129-5p, -133a, and -215) are downregulated [139]. Results of a study on Merkel cell carcinoma revealed that serum levels of miRNA-375 directly correlated with tumor burden during disease progression in patients treated with therapeutic interventions (radiation therapy, chemotherapy, and immunotherapy) [142]. In addition, high levels of miR-375 in both tissues [143] and plasma [144] of patients with prostate NEC have been correlated with poor overall survival [144]. These studies suggest further investigation into the potential role of miR-375 as a biomarker for monitoring and treatment management. Moreover, the levels of miRNA-181b-5p and miRNA-181a-2-3p are correlated with the efficiency of surgery in patients affected by pituitary NETs secreting GH [145]. In the plasma, the miRNA-181b-5p was upregulated 24 h after surgery and downregulated in GH-secreting patients compared to non-functioning pituitary NET patients, whereas miRNA-181a-2-3p was upregulated in GH patients 24 h after surgery and downregulated in GH patients before surgery compared to non-functioning patients before surgery [145].

The presence of miRNAs might be dependent on the stage, the metastatization status, and the treatment status of the patients’ sample [146,147,148]. However, to date, few studies investigated the role of miRNAs in detecting early relapse in NENs. In pancreatic NETs, the increase in miRNA-183-5p, miRNA132-3p, miRNA 145-5p, miRNA34a-5p, and miRNA 449a were associated with a worse prognosis [149,150]; other groups identified miRNA-210 as a potential prognostic biomarker of metastatization in pancreatic NETs [150,151,152,153]. In patients with small intestinal NETs, the increase in miRNA-200a was associated with metastasis formation in both untreated and SSA-treated patients, whereas its levels were normal in the earlier stages of the disease [139]. 

Although miRNAs may be a promising tool, to date, this field of research is still at an early stage. It would be necessary to stimulate the study of the roles of miRNAs in specific types and grades of NENs combined with a deep clarification of underlying mechanisms.

#### 3.2.3. Circulating Tumor DNA 

The circulating tumor DNA (ctDNA) fragments are composed of approximately 150 bp and derived from apoptotic, necrotic, and autophagic processes [12]. ctDNA is present in body fluids as free, protein-bound, or in extracellular vesicles and can be clinically detected by non-invasive methods, including liquid biopsy [152]. Since these molecules are derived from cancer cells, ctDNAs can carry the genetic and epigenetic mutation profile of the tumor of origin. Moreover, ctDNAs are characterized by rapid turnover, making it possible to monitor tumor evolution in real time [58,73,154,155,156,157,158,159,160,161]. 

Despite the encouraging perspectives, the research field on the diagnostic role of ctDNA in NENs is still at the beginning (Table 3). High levels of ctDNA differentiated pancreatic and small intestine NET from healthy controls [157]. Recently, ctDNA concentration has been correlated with high grade and proliferation index associated with metastasis in the liver, typical of NECs [157,158]. In particular, the correlation of high grade has been reported to be able to differentiate metastatic and localized pancreatic NETs [157]. Indeed, the lack of knowledge of the mutational profile characterizing the different subtypes of NENs and the low value of tumor mutation burden for most of them make their use limited [59,162].

Quantitative analysis of ctDNA may be useful to assess tumor volume as a predictive factor of response to treatment [59,163]. Indeed, a reduction in ctDNA associated with a longer PFS was reported in patients with lung and GEP-NETs treated with everolimus [157]. In pancreatic NETs, the increase in mutations and CNAs found in plasma ctDNAs during the follow-up showed a correlation with higher tumor burden and tumor progression [157]. This promising result was also confirmed in another study to assess treatment response in patients with metastatic GEP or of unknown origin NEC treated with chemotherapy [158]. Moreover, in two case reports on Merkel cell carcinoma, ctDNA levels were found to be correlated with tumor burden and response to treatment [164]. ctDNA measurement was used to track the disease course in two patients with Merkel cell carcinoma whose disease had progressed on pembrolizumab, a programmed cell death-1 (PD-1) inhibitor [165]. Interestingly, ctDNA changes can also be measured in urine to monitor therapy, as was demonstrated in a case of metastatic high-grade rectal NET refractory to treatment [29].

The detection of new clonal mutations in ctDNA derived from patients with NENs could pave the way as a tool to trace the tumor relapse [158,159]. In 18 patients with NEN during everolimus treatment, a joint modeling approach showed a significant association between longitudinal tumor fraction measurements in ctDNA and the risk for tumor recurrence [157]. Indeed, higher tumor fractions could be observed before disease progression, while a decreasing tumor fraction could be observed in patients with durable stable disease. The methylome profiling of ctDNA showed clinically relevant methylation signatures involved in tumor progression from serum or plasma of patients with pituitary NETs identified also in tissue samples [166]. The ctDNA integrity and the hypomethylation status of repetitive DNA sequences like *Alu* or *LINE-1* in ctDNA have recently been evaluated as an attractive non-invasive biomarker to evaluate both tumor diagnosis and relapse in various types of cancers [167]. Indeed, higher ctDNA concentration with a stronger global *Alu* hypomethylation and reduced *LINE-1* integrity were found in the plasma of patients with metastatic NENs compared with patients with localized NENs or healthy control group [167]. These parameters were strongly associated with tumor burden, without the correlation with tumor localization, hormonal activity, or mitotic activity [167]. This study suggests that ctDNA hypomethylation levels combined with plasma ctDNA concentration and integrity may be a useful non-invasive biomarker to detect recurrent or metastatic disease, the prognosis of the patients, and treatment response.

**Table 3 diagnostics-13-02820-t003:** Strengths and flaws of multi-analyte biomarkers in early diagnosis, monitoring response to therapy, and early detection of tumor relapse.

Multi-Analyte Biomarkers	Strengths	Flaws
NETest	High diagnostic accuracy in NENs (>90%) [115,116,168]	Not standardized cut-off values to distinguish stable from progressive disease [72]
	Able to differentiate stable (score < 40%) from progressive disease in NENs [72,116,117]	Specificity influenced by the presence of gastrointestinal tract benign diseases in GEP-NET [117,118]
	Able to monitor response to therapy in GEP-, bp NET and of unknown origin [72,117,121]	
	Able to tumor recurrence after surgery (score < 33–40%) in NEN [123,124,126]	
miRNA	Able to discriminate NET from carcinoma and benign disease in the pNET and siNET [134,138,139]	Different expression between tumor tissue and body fluids for the same miRNA [136]
	Correlation between expression variation and tumor progression in different NENs [139,141,142,144]	Lack of standardization guidelines for analysis methods [136]
	Correlation between expression variation metastatization and worse prognosis [149,150,151]	Not evaluable in G1 and G2 NET [111,143]
ctDNA	Able to discriminate pNET and siNET from healthy controls [157]	Limited diagnostic value in low tumor burden NENs [59,162]
	Able to discriminate metastatic and localized pancreatic NETs [157]	
	Variations in the amount predicts PFS in lung and GEP-NET [157]	
	Mutations and CNAs detected are useful to predict response to treatment in GEP-NET, NEC, and Merkel cell carcinoma [157,158]	
	Methylome profile obtained is able to identify tumor progression and evaluates presence of metastasis in NENs [164]	

miRNAs = microRNAs; ctDNA = circulating tumor DNA; GEP = gastroenteropancreatic; PFS= progression-free survival; OS = overall survival; CNAs = copy number alterations; bp NET = broncopulmonary-NET; pNET = pancreatic NET; siNET = small intestinal NET.

## 4. Conclusions and Future Directions

Despite the huge effort in finding robust biomarkers for NEN, there is still an urgent need to develop biomarkers that meet diagnostic accuracy combined with driving therapeutic options and tracing the early relapses. To date, mono-analytes have different limitations due to the inability to describe the complexity of NENs [17]. On the other hand, a common opinion is that the use of a multi-analytical panel will be essential in diagnosing NENs [21]. 

CgA showed an acceptable sensitivity only for well-differentiated NENs and may predict disease progression, especially in advanced NENs and gastrinomas. However, the absence of a CgA-specifc standardized test and poor specificity, raising serious concerns about its potential clinical use (Table 1 and Table 3).

The increase in CTCs is more indicative of metastasis formation and chemo-resistance, but they are not always detectable (<50% of NENs) (Table 1 and Table 3).

Differently, miRNA and ctDNA may predict tumor progression and metastasis formation with a high sensitivity but they lack standardization guidelines and their diagnostic applicability in NEN is still limited (Table 1 and Table 3). 

To date, the NETest showed better performance for early diagnosis, monitoring of therapeutical efficiency, and detection of tumor relapse (Table 1 and Table 3).

Notably, a consideration of the cost–benefit ratio is also relevant. On the one hand, mono-analyte biomarkers are low-cost assays with controversial specificity, while, multi-analyte biomarkers are expensive assays with higher specificity.

Overall, a direct comparison of these two different types of biomarkers through a larger number of comparative studies, including multi-institutional studies, would offer valuable insights about their validity as clinically diagnostic tools. In particular, a comparison between NETest versus currently used biomarkers, such as CgA and imaging tools might ensure the accuracy of the NETest compared to mono-analyte biomarkers, providing robust evidence for its use in diagnostics.

## Figures and Tables

**Figure 1 diagnostics-13-02820-f001:**
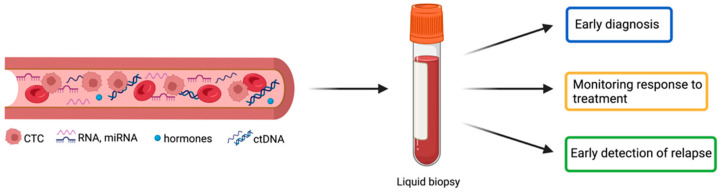
The perspective of liquid biopsy in early diagnosis, monitoring of therapeutical efficiency, and detection of tumor relapse in NENs. NENs = neuroendocrine neoplasms; CTC = circulating tumor cell; miRNA = microRNA; ctDNA = circulating tumor DNA.

## Data Availability

Not applicable.

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
