# Peer review of "An Overview of Circulating Biomarkers in Neuroendocrine Neoplasms: A Clinical Guide"

_diagnostics, 2023, doi:10.3390/diagnostics13172820_

Round 1

Reviewer 1 Report

Bevere et al.  aim to summarize in their review the most innovative biomarkers used in diagnosis and monitoring of neuroendocrine neoplasms. The details presented on the described biomarkers are well documented, however there are some major issues with the contents of this review.

The main problem with this review is that it is not clear if the authors wanted to formulate a description of the circulating biomarkers or to include all the biomarkers used in NENs, as the title suggests. The authors state in the abstract and lines 58-59 that we will analyze the capabilities and limitations of different biomarkers focusing on 20 three relevant questions: (1) accurate and early diagnosis; (2) monitoring of disease progression and 21 response to therapy; and (3) detection of early relapse” and This state-of-the-art review aims to summarize the most innovative biomarkers and 58 diagnostic tools which may lead to an easier and faster diagnosis of NENs.”, however major biomarkers are missing completely. I suggest revising the contents of this review, adding some missing circulating biomarkers, such as calcitonin in the case of medullary thyroid carcinomas, catecholamines and their metabolites for pheochromocytomas,  neuropeptide Y for pargangliomas etc. Furthermore, no description of the immunohistochemical markers has been included in this review, knowing that some of them may represent ways of diagosis and treatment monitoring (such as SSTR expression). Some other novel biomarkers, that have been studied in the past years, are also missing, such as different genes expression (DAXX, ATRX, RB1), VCAM-1, VGEF, INSM-1, to name a few. I suggest adding them in your article, because numerous other papers on the markers analysed in your review have already been published and this paper does not bringing any novelty on this topic.

I suggest reading these articles for a better guidance on the subject: PMID: 36233409, PMID: 34765020, PMID: 36135186

There are also some problems with the structure of the paper:

  1. There is no section 2
  2. I suggest adding the search algorithm for the references included in this paper (such as used databases, search key words etc.)
  3. Add a paragraph describing what mono- and multianalyte biomarkers are and the differences between the two.
  4. Once again, pay attention to the naming and numbering of the sections and sub sections (i.e. subsection 1.2 is under section 3 etc.)

I suggest thoroughly revising the paper and including more data on this topic, to make it suitable for publication.

 I suggest a reevaluation of some of the topic of English language. I have not detected major language errors, but I believe that some phrases can be reformulated. 

Author Response

Reviewer 1

Bevere et al.  aim to summarize in their review the most innovative biomarkers used in diagnosis and monitoring of neuroendocrine neoplasms. The details presented on the described biomarkers are well documented, however there are some major issues with the contents of this review.

The main problem with this review is that it is not clear if the authors wanted to formulate a description of the circulating biomarkers or to include all the biomarkers used in NENs, as the title suggests. The authors state in the abstract and lines 58-59 that “we will analyze the capabilities and limitations of different biomarkers focusing on 20 three relevant questions: (1) accurate and early diagnosis; (2) monitoring of disease progression and 21 response to therapy; and (3) detection of early relapse” and “This state-of-the-art review aims to summarize the most innovative biomarkers and 58 diagnostic tools which may lead to an easier and faster diagnosis of NENs.”, however major biomarkers are missing completely. I suggest revising the contents of this review, adding some missing circulating biomarkers, such as calcitonin in the case of medullary thyroid carcinomas, catecholamines and their metabolites for pheochromocytomas, neuropeptide Y for pargangliomas etc. Furthermore, no description of the immunohistochemical markers has been included in this review, knowing that some of them may represent ways of diagosis and treatment monitoring (such as SSTR expression). Some other novel biomarkers, that have been studied in the past years, are also missing, such as different genes expression (DAXX, ATRX, RB1), VCAM-1, VGEF, INSM-1, to name a few. I suggest adding them in your article, because numerous other papers on the markers analysed in your review have already been published and this paper does not bringing any novelty on this topic. I suggest reading these articles for a better guidance on the subject: PMID: 36233409, PMID: 34765020, PMID: 36135186.

We appreciate the reviewer’s suggestion and advice. First, we clarified the aim of the review focusing on the main circulating biomarkers with clinical relevance in the field of NENs. Accordingly, we modified the title with the addition of “circulating” and we described more clearly the aim of this review in the introduction (Lines 60-61). Furthermore, we included other circulating biomarkers as suggested, such as calcitonin in the case of medullary thyroid carcinomas, catecholamines and their metabolites for pheochromocytomas, and neuropeptide Y for paragangliomas in the chapter “Other biomarkers” (3.1.3) (Lines 202-326). At the end of this chapter, we included a table (Table 2) summarizing each tumor-specific biomarker used for the diagnosis of different functioning NENs.

There are also some problems with the structure of the paper:

  1. There is no section 2
  2. I suggest adding the search algorithm for the references included in this paper (such as used databases, search key words etc.)
  3. Add a paragraph describing what mono- and multianalyte biomarkers are and the differences between the two.
  4. Once again, pay attention to the naming and numbering of the sections and sub sections (i.e. subsection 1.2 is under section 3 etc.)

I suggest thoroughly revising the paper and including more data on this topic, to make it suitable for publication.

We changed it as suggested. We included the description of the search algorithm for the references at the end of introduction (Lines 63-72) and section 2 with a brief description of the mono- and multianalyte biomarkers with their differences (Lines 73-82).

Comments on the Quality of English Language

I suggest a reevaluation of some of the topic of English language. I have not detected major language errors, but I believe that some phrases can be reformulated. 

According to the suggestion, we revised the English language of the whole manuscript with the aid of a native English member of our team.

Reviewer 2 Report

Well rounded, comprehensive and on topic review.

Author Response

Reviewer 2

Well rounded, comprehensive and on topic review.

We really thank you for this comment.

Reviewer 3 Report

In this paper a review of biomarkers in neuroendocrine tumors is presented.

It is a review of a very broad subject, of a very heterogeneous group of tumors due to their varied origin.

The authors give more information on CgA and CTC than the other markers (3.2) where the information is very short. The review of NETest, MicroRNAs, and ctDNA is current and complete.

 In section 1.2 on line 133, reference 52 does not correspond to the text

Section 3.2 line 183 reference 74 is very old, it is recommended to change it for a recent one

Section 3.2.2 line 340 reference 127 does not correspond

Author Response

Reviewer 3

In this paper a review of biomarkers in neuroendocrine tumors is presented.

It is a review of a very broad subject, of a very heterogeneous group of tumors due to their varied origin.

The authors give more information on CgA and CTC than the other markers (3.2) where the information is very short. The review of NETest, MicroRNAs, and ctDNA is current and complete.

Thank you for the suggestion. We added a section named “Other biomarkers” (3.1.3) to give a summary of other circulating biomarkers (Lines 202-326).

 In section 1.2 on line 133, reference 52 does not correspond to the text

Section 3.2 line 183 reference 74 is very old, it is recommended to change it for a recent one

Section 3.2.2 line 340 reference 127 does not correspond

We appreciate the reviewer’s recommendation. We replaced references #52 and #127 with appropriate references (Section 3.1.2, lines 161, reference #55; Section 3.2.2, lines 485, reference #158; respectively). Moreover, we changed the old reference 74 with a recent one (Section 3.1.3, lines 263, reference #90).

Reviewer 4 Report

In the manuscript “An overview of biomarkers in neuroendocrine neoplasms: a clinical guide”, the authors present and discuss the current challenges in the diagnosis and follow-up of neuroendocrine neoplasms using available biomarker tests.

It would enrich the manuscript to further discuss data related to biomarkers mentioned at lines 168 & 169. Lines 170-186 were too superficial and deserve more in-depth discussion.

Unnecessary self-citations: Reference #1 and 2. Switch to a comprehensive reference.

Suggestions/comments:

Line 13: include “are” after which

Line 34: Include the description of the abbreviations of NEC and NET.

Line 50: current “European” guidelines, otherwise include NCCN reference(#38). Include the last updated reference, there were updates since 2020.

Line 91: exemplify common confounding factors.

Line 107: Include the description of the abbreviation of PRRT

Line 126-129: Important that you had mentioned that.

Line 140-142: same as above.

Line 147: Include the description of the abbreviation of SSA

Line 165: “in patients with”, rephrase it.

Line 166: Correct “3.2”

Line 176: describe the triad.

Line 190: “disease”

Line 225: “(<10000 samples)”, be specific.

Line 249: “polmunary”

Line 283: “detetction”

Line 287: stage IV

Line 334-337: reword “GH patients’ plasma”

Line 373: “This promising results was”

Line 405: “gastroenteropancratic”

Add adequate reference formats:

#38: This link is related to Acute Lymphoblastic Leukemia. Switch to: “Network, N.C.C., Neuroendocrine and Adrenal Tumors, Version 2.2022” or to the 2021 version “Shah MH, Goldner WS, Benson AB, Bergsland E, Blaszkowsky LS, Brock P, Chan J, Das S, Dickson PV, Fanta P, Giordano T, Halfdanarson TR, Halperin D, He J, Heaney A, Heslin MJ, Kandeel F, Kardan A, Khan SA, Kuvshinoff BW, Lieu C, Miller K, Pillarisetty VG, Reidy D, Salgado SA, Shaheen S, Soares HP, Soulen MC, Strosberg JR, Sussman CR, Trikalinos NA, Uboha NA, Vijayvergia N, Wong T, Lynn B, Hochstetler C. Neuroendocrine and Adrenal Tumors, Version 2.2021, NCCN Clinical Practice Guidelines in Oncology. J Natl Compr Canc Netw. 2021 Jul 28;19(7):839-868. doi: 10.6004/jnccn.2021.0032. PMID: 34340212.”

#103: “Pavel M, Jann H, Prasad V, Drozdov I, Modlin IM, Kidd M. NET Blood Transcript Analysis Defines the Crossing of the Clinical Rubicon: When Stable Disease Becomes Progressive. Neuroendocrinology. 2017;104(2):170-182. doi: 10.1159/000446025. Epub 2016 Apr 15. PMID: 27078712.”

Moderate editing of English language required, special note to multiple typos.

Author Response

Reviewer 4

In the manuscript “An overview of biomarkers in neuroendocrine neoplasms: a clinical guide”, the authors present and discuss the current challenges in the diagnosis and follow-up of neuroendocrine neoplasms using available biomarker tests.

It would enrich the manuscript to further discuss data related to biomarkers mentioned at lines 168 & 169. Lines 170-186 were too superficial and deserve more in-depth discussion.

We really appreciate the reviewer’s suggestion. We included more details in the chapter “Other biomarkers” (3.1.3) to describe each biomarker used in different functioning NENs Lines (Lines 202-326).

Unnecessary self-citations: Reference #1 and 2. Switch to a comprehensive reference.

We replaced references #1 and 2 with a comprehensive reference for the definition of neuroendocrine neoplasms (NENs).

Suggestions/comments:

Line 13: include “are” after which

Line 34: Include the description of the abbreviations of NEC and NET.

Line 50: current “European” guidelines, otherwise include NCCN reference(#38). Include the last updated reference, there were updates since 2020.

Line 91: exemplify common confounding factors.

Line 107: Include the description of the abbreviation of PRRT

Line 126-129: Important that you had mentioned that.

Line 140-142: same as above.

Line 147: Include the description of the abbreviation of SSA

Line 165: “in patients with”, rephrase it.

Line 166: Correct “3.2”

Line 176: describe the triad.

Line 190: “disease”

Line 225: “(<10000 samples)”, be specific.

Line 249: “polmunary”

Line 283: “detetction”

Line 287: stage IV

Line 334-337: reword “GH patients’ plasma”

Line 373: “This promising results was”

Line 405: “gastroenteropancratic”

We thank you for these suggestions/comments. We modified each point as indicated.

Add adequate reference formats:

#38: This link is related to Acute Lymphoblastic Leukemia. Switch to: “Network, N.C.C., Neuroendocrine and Adrenal Tumors, Version 2.2022” or to the 2021 version “Shah MH, Goldner WS, Benson AB, Bergsland E, Blaszkowsky LS, Brock P, Chan J, Das S, Dickson PV, Fanta P, Giordano T, Halfdanarson TR, Halperin D, He J, Heaney A, Heslin MJ, Kandeel F, Kardan A, Khan SA, Kuvshinoff BW, Lieu C, Miller K, Pillarisetty VG, Reidy D, Salgado SA, Shaheen S, Soares HP, Soulen MC, Strosberg JR, Sussman CR, Trikalinos NA, Uboha NA, Vijayvergia N, Wong T, Lynn B, Hochstetler C. Neuroendocrine and Adrenal Tumors, Version 2.2021, NCCN Clinical Practice Guidelines in Oncology. J Natl Compr Canc Netw. 2021 Jul 28;19(7):839-868. doi: 10.6004/jnccn.2021.0032. PMID: 34340212.”

#103: “Pavel M, Jann H, Prasad V, Drozdov I, Modlin IM, Kidd M. NET Blood Transcript Analysis Defines the Crossing of the Clinical Rubicon: When Stable Disease Becomes Progressive. Neuroendocrinology. 2017;104(2):170-182. doi: 10.1159/000446025. Epub 2016 Apr 15. PMID: 27078712.”

We modified the reference format, as suggested.